# Study of the Agglomeration Characteristics of Cultivated Land in Underdeveloped Mountainous Areas Based on Spatial Auto-Correlation: A Case of Pengshui County, Chongqing, China

Guanglian Luo [1], Bin Wang [2], Ruiwei Li [3], Dongqi Luo [4] and Chaofu Wei [5,*]

1   College of Public Administration, Chongqing Technology and Business University, Chongqing 400067, China; lglzyx@163.com
2   Chongqing Zhiheng Land Planning and Design Co., Ltd., Chongqing 400067, China; wangbinchqzlpd@163.com
3   College of Business Administration, Chongqing Technology and Business University, Chongqing 400067, China; 2021335087@email.ctbu.edu.cn
4   Research Center of the Economy of the Upper Reaches of the Yangtze River of the Key Research Base of Humanities, Ministry of Education, Chongqing Technology and Business University, Chongqing 400067, China; luodongqi@ctbu.edu.cn
5   College of Resources and Environment, Southwest University, Chongqing 400715, China
*   Correspondence: weicf@swu.edu.cn; Tel.: +86-135-0941-9507

**Abstract:** The economic and social orientation of cultivated land in underdeveloped mountainous areas is obvious. A study of the spatial agglomeration characteristics of cultivated land quality can provide guidance for regional economic and social development. Taking Pengshui County, Chongqing, China as the study area, the spatial agglomeration characteristics of cultivated land quality indexes at county, township and village levels were analyzed by using the auto-correlation analysis method. The results showed that: (1) At different spatial scales, the cultivated land quality index showed spatial agglomeration characteristics. (2) Moran's I values of the cultivated land quality index at county, township and village level decreased successively, but three indexes still showed significant positive spatial correlation. (3) The spatial scale affects the spatial agglomeration of the cultivated land quality index, and its influence is physical, with a utilization and economic quality grade index from large to small. In underdeveloped mountainous areas, the spatial agglomeration characteristics of township scale and physical quality grade index are the most stable and significant, which can be used as the direct basis for zoning of cultivated land protection and site selection of rural residents' agglomeration points.

**Keywords:** underdeveloped mountainous areas; spatial agglomeration; cultivated land quality index; spatial auto-correlation; space scale

## 1. Introduction

Compared with plain areas, mountainous areas are generally poor and highly dependent on the natural environment, especially cultivated land. The mountainous areas on Earth account for about 24% of the total land area, and mountainous areas in China account for about 2/3 of the total land area [1], which is a region with lagging economic development [2]. Cultivated land is an important production, living and ecological space in underdeveloped mountainous areas, an important material basis for the sustainable development of regional economy and society [3], and an important strategic resource for the country. At the same time, the economic and social orientation of underdeveloped mountainous areas is more obvious [4,5]. It is of great theoretical and practical significance to study the spatial agglomeration characteristics of cultivated land quality in mountainous areas.

Academic circles have studied the spatial agglomeration of cultivated land in mountainous areas. Perhaps because the proportion of mountainous areas in China is much higher than the world average, there are relatively few regional studies outside China. In a study of soil microbial community in the mountainous area of eastern Spain mentions that economic and social changes; orchards were abandoned, and special agricultural land was becoming concentrated [6]. Natural forests or productive forests in Vietnam are also mentioned as increasing the use of irrigated terraces to concentrate cultivated land to compensate for converting land into forests [7]. In the central Highlands of Ethiopia, the concentration of human and livestock populations has led to changes in land use and land cover, especially arable land [8]. In a study on the impact of agricultural planting on soil and water loss in hilly and mountainous areas, it was mentioned that terraced fields were traditionally adopted and that heavy modern mechanization was concentrated on flat land [9]. In a study of soil erosion in the Carpathian mountains in western Poland, soil erosion was mentioned as one of the most important processes affecting land degradation in mountainous areas, and topography dominates the concentration of cultivated land [10]. Research on the marginalization of arable land and its driving factors in mountainous areas of China refers to the shift from labor-intensive to machine-intensive with the declining profits of arable land. Cultivated land in mountainous areas gathers and makes up for the loss of cultivated land by improving the quality and utilization efficiency of surplus cultivated land [11]. In an evaluation study on the impact of land segmentation on agricultural operation costs in mountainous areas of southwest China, it was proposed that in order to reduce the land system costs in mountainous areas, the government should reasonably integrate land, achieve moderate scale operations, reduce the transaction costs of land transfers, and promote machinery suitable for complex terrain [12]. A study of average family size in different regions of Shawan City, Xinjiang, China, found that the average suitable operating scales of households in the hilly areas, the agricultural area in the oasis plain, and the oasis–desert interlace area were 5.15, 9.28, and 7.74 ha, respectively [13]. In the hilly and gully region of the Loess Plateau of China, a study on the change of abandoned cultivated land noted that the finely fragmented and dispersed cultivated land had been abandoned. The higher the slope of cultivated land is, the lower the soil nutrient content is, and the more likely it is to be abandoned [14]. In Sichuan, China, a study of the spatial and temporal differentiation of cultivated land in complex geomorphic areas and its response to climate factors mentions that cultivated land is mainly distributed in low-altitude hills and low-altitude hilly areas, and that the maximum temperature offers the strongest explanation for the spatial heterogeneity of cultivated land [15]. In Jiangsu province, China, a study on the evolution of cultivated land refinement and its driving mechanism for rural development suggests that the land area should be expanded to integrate with rural residential space in order to alleviate the refinement of cultivated land [16]. In a study on the impact of land use transformation on the ecological vulnerability of poverty-stricken mountainous areas in 16 counties in western Hubei Province, China, the conversion of concentrated agricultural land into construction land is identified as the main change feature, and that it has a negative impact on ecology [17].

Research on spatial agglomeration methods of cultivated land has the following literature. In an analysis of cultivated land fragmentation and spatial agglomeration pattern in Jiaxing, China, spatial auto-correlation was adopted to explore the coupling relationship between cultivated land fragmentation and regional socio-economic spatial patterns [18]. In the analysis of the characteristics of land use transfer flow and spatial agglomeration in the Loess tableland of China, density mapping was applied to the identification of spatial agglomeration characteristics of land use change [19]. Based on the spatial agglomeration pattern and boundary modification, the regional Moran's I index was used to determine the spatial agglomeration pattern of high-quality prime farmland [20]. In the multi-scale spatial auto-correlation analysis of cultivated land quality in Zhejiang Province, China, the spatial auto-correlation based on Moran's I index was used to analyze the spatial characteristics of cultivated land quality at different scales, so as to carry out targeted construction

of cultivated land quality and formulate reasonable management measures [21]. The spatial measurement system of cultivated land fragmentation was constructed from three resource elements: scale; spatial agglomeration; and production convenience The multi-dimensional evaluation of cultivated land fragmentation at provincial scale in Jiangsu Province, China, and the corresponding collaborative discussion of land consolidation were carried out by using geospatial analysis and mathematical statistical analysis [22]. The local spatial auto-correlation method was used to quantitatively characterize the spatial agglomeration pattern characteristics of cultivated land quality in Jinan, China, and the matrix grouping method was used to further combine the cultivated land quality grade and spatial agglomeration pattern types in pairs to comprehensively determine the basic farmland delineation type [23]. Intensive use of cultivated land is crucial to optimize crop planting methods and protect food security, and remote coupling frameworks are used to evaluate the intensification rate of cultivated land use in typical villages in China [24]. Accurate identification of cultivated land quality is the prerequisite to ensure food security, and spatial analysis and multi-factor comprehensive evaluation were used to evaluate the spatial distribution characteristics of cultivated land quality in alluvial fan terrain in the arid region of Jimsar County, Xinjiang, China from the three dimensions of soil properties, farming conditions, and natural environment conditions [25].

Other scholars have also conducted similar studies [26–36]. Existing studies have paid little attention to the effects of different scales on the spatial agglomeration of cultivated land, which may weaken its application value. In this paper, spatial auto-correlation was used to analyze the spatial agglomeration characteristics of cultivated land quality in developed mountainous areas at the spatial scale of county, township and village, providing evidence for the protection of cultivated land use and reconstruction of rural agglomeration points.

## 2. Case Study

### 2.1. Overview of Research Region

Pengshui County is located in the southeast of Chongqing, China, in the lower reaches of Wujiang River, a tributary of the Yangtze River, between E 07°48′–108°35′, N 28°57′–29°50′. The county is 77.88 km wide from east to west and 96.40 km long from north to south, covering a total area of 3903 km$^2$. It has jurisdiction over 3 subdistricts, 18 towns and 18 townships, with a total of 296 villages. Wuling mountain system, the territory of the northwest high, southeast low, landform "two mountains sandwich a trough"; Hills and valleys, low mountains and middle mountains account for 13.39%, 52.88% and 4.03%, respectively. Pengshui County has a highest elevation of 1859.60 m, a lowest elevation of 190 m, and a relative elevation difference of 1669.60 m, which is the middle and low mountain terrain of tectonic denudation. The Wujiang River runs from the southeast to the west across the territory, a length of 64 km; There is also the Yujiang River, Changxi River, Furong River, Wood Palm River and more than 20 other tributaries (Figure 1). Pengshui County has a subtropical humid monsoon climate: mild climate, abundant rainfall, annual average temperature of 17.5 °C, annual average precipitation of 1228.7 mm, annual sunshine duration of 1035.5 h, and a frost-free period of 311 days. The soil is mainly subtropical mountain yellow soil. At the end of 2020, Pengshui County had a total population (household registration) of 703,000, urban population of 206,600 (29.39% of the total population), and rural population of 496,400 (70.61%). It is a typical mountain agricultural county with Miao nationality as the main minority. The GDP of the county is CNY 24.51 billion, ranking 34th among 39 districts (counties) in Chongqing. The per capita disposable income of permanent rural residents is RMB 11,620, ranking 36th among 39 districts (counties) in Chongqing, lower than the city's average level (CNY 16,361), and the economy is less developed [37].

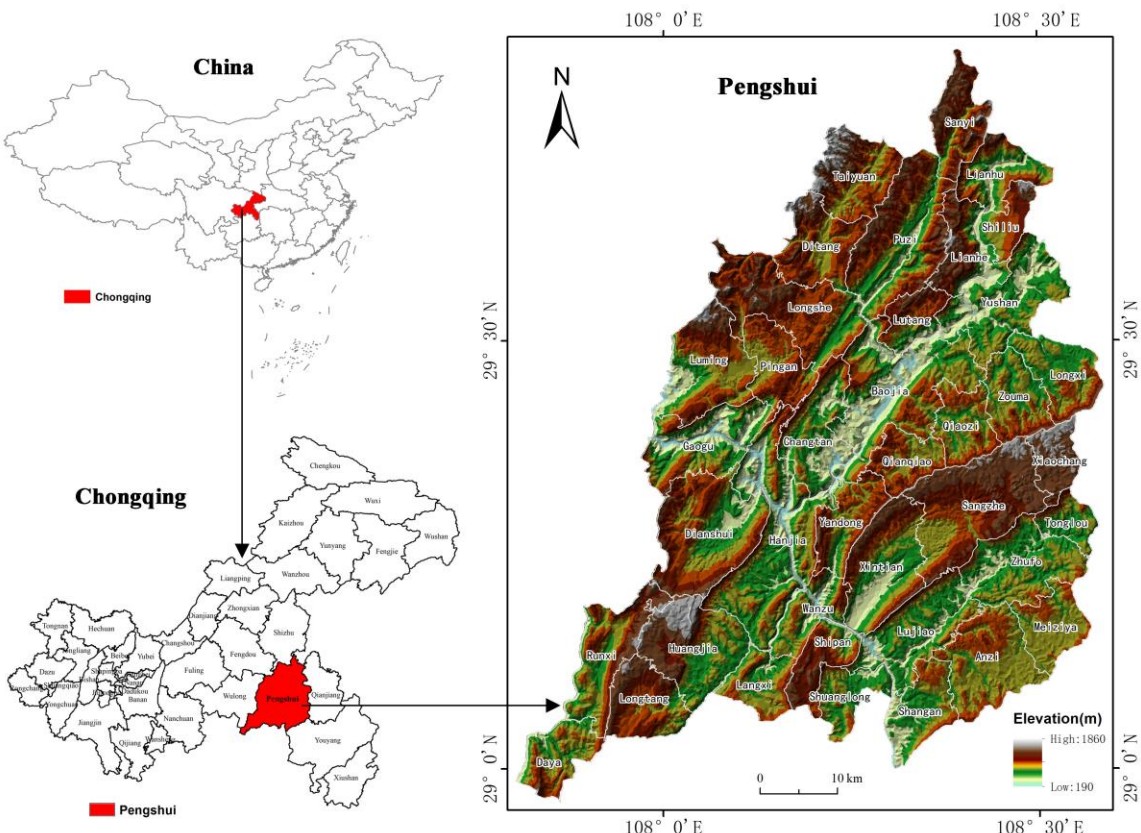

**Figure 1.** Location of the study area.

Compared with the previous year, in 2020, the annual sown area of grain in Pengshui county was 77,700 ha, down 0.30%; the sown area of oil was 17,900 ha, up 3.80%; the sown area of flue-cured tobacco was 40,400 ha, down 4.20%; and the sown area of vegetables was 16,300 ha, up 3.80%. Total grain output for the year was 312.20 million kg, up by 0.40%. Among them, the output of summer grain was 73.40 million kg, down by 0.50%. The output of autumn grain was 238.80 million kg, up by 0.70%. Total oil production was 33.30 million kg, up by 4.80%. Total flue-cured tobacco output was 8.00 million kg, down by 4.40%. The total output of vegetables was 426.60 million kg, up by 3.80%. In 2020, the output of main agricultural products in Pengshui County is shown in (Table 1).

**Table 1.** The output of major agricultural products in Pengshui County in 2020.

| Product Name | Production (Million kg) | Last Year ($\pm$%) |
|---|---|---|
| Food | 312.20 | +0.40 |
| Corn | 112.50 | +0.20 |
| Rice | 63.00 | +0.30 |
| Potato | 128.30 | +0.50 |
| Oil | 33.30 | +4.80 |
| Vegetables | 426.60 | +3.80 |
| Flue-cured tobacco | 8.00 | −4.40 |
| Fruit | 9.80 | +3.00 |

## 2.2. Data Source and Processing

The research data mainly came from the administrative division, land use status database, cultivated land quality database, and statistical Yearbook of Pengshui County in 2020 [38]. Distribution of cultivated land in Pengshui County (Figure 2).

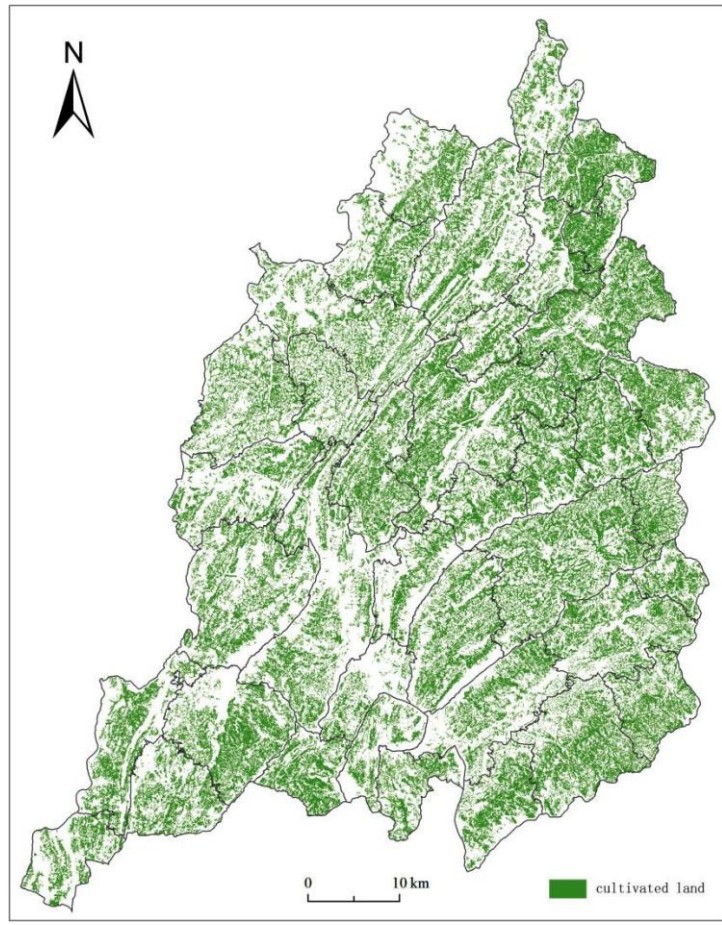

**Figure 2.** Distribution of cultivated land of study area.

The physical condition, utilization condition, and economic level of cultivated land form the important bases to measure the quality of cultivated land. Physical, utilization, and economy of quality grade index are usually used to describe the quality of cultivated land. Among them, physical quality grade index is the comprehensive reflection of the natural property of cultivated land. Utilization quality grade index is the product of the ratio of the designated crop yield to the highest yield per unit area in the province and the physical index, which is a comprehensive reflection of the cultivated land output efficiency in this region. Economic quality grade index is the product of the ratio of the actual "yield-cost" index of a specified crop and the highest "yield-cost" index in the province of the crop and the utilization index, which is a comprehensive reflection of the input and output efficiency of cultivated land in this region [38].

## 3. Research Methods

### 3.1. Spatial Weight

Appropriate spatial weight is the basis of spatial auto-correlation research. The spatial weights mainly include adjacent weight, distance weight, and nearest K point weight. Since the plot spots of cultivated land are not connected to edges, the weight of the nearest K point is adopted. The spatial weight value refers to the spatial adjacency frequency histogram based on the Queen and Rook principles of adjacency weight, and the number of adjacent spatial units in the study area "4" is selected for spatial auto-correlation analysis [39].

### 3.2. Spatial Auto-Correlation

According to the first law of geography, spatial auto-correlation between things exists objectively, and the degree of aggregation or dispersion of attribute values of spatial

elements between things can be measured by global spatial auto-correlation and local auto-correlation index. Based on Arc GIS and Geo Da platform, this paper uses global Moran's I index and local Moran's I index to analyze the spatial dependence of cultivated land quality index in Pengshui County at county, township, and village level. The relevant calculation formula is as follows [40].

Global Spatial Auto-Correlation:

$$I = \frac{n \sum\limits_{i=1}^{n} \sum\limits_{j=1}^{n} w_{ij} \times (x_i - \overline{x}) \times (x_j - \overline{x})}{\left( \sum\limits_{i=1}^{n} \sum\limits_{j=i}^{n} W_{ij} \right) \sum\limits_{i=1}^{n} (x_i - \overline{x})^2}$$

Local Spatial Auto-correlation:

$$I = \frac{x_j - \overline{x}}{\sqrt{\sum\limits_{i=1}^{n} (x_i - \overline{x})^2 / (n-1)}} \sum\limits_{j=1}^{n} w_{ij}(x_j - \overline{x})$$

In formula, $n$ represents the number of research objects; $x_i$ is the observed value, and $\overline{x}$ is the average value of $x_i$; $W_{ij}$ is the spatial connection matrix between spatial elements $i$ and $j$; $x_j$ represents the attribute value in the $j$ region, and $\overline{x}$ represents the average value of the attribute value in the studied region; $W_{ij}$ represents the spatial weight matrix, which is generally symmetric. Moran's I value is between $(-1,1)$, and $I > 0 > 0$ indicates positive spatial correlation, and the elements tend to be spatially aggregated. $I < 0$ indicates negative spatial correlation, and the elements tend to be spatially discrete. $I = 0$ represents random analysis of spatial elements, and $p$ value is generally used for significance test.

### 3.3. Variable Coefficient

Variable coefficient is the ratio between standard deviation and average value, which reflects the degree of agglomeration or dispersion and is not affected by the measurement scale and dimension of the research variable [41]. In order to explore the relationship between scale change and spatial auto-correlation of cultivated land quality, the coefficient of variation was used to measure the internal difference of spatial auto-correlation of cultivated land quality at different spatial scales. The specific calculation formula is as follows:

$$C_v = \sqrt{\sum\limits_{f=1}^{n} (y_f - \overline{y}) / (n-1) / \overline{y}}$$

In formula, $C_v$ is the coefficient of variation; $n$ is the total number of corresponding units; $y_f$ is Moran's I value of cultivated land quality in unit $f$; $\overline{y}$ is the average of $y_f$.

## 4. Results

### 4.1. Overall Spatial Distribution Characteristics of Cultivated Land Quality

The cultivated land grade data were extracted from the achievements database of cultivated land quality in Pengshui County in 2020 and classified for statistics (Table 2). In terms of area, the cultivated land in Pengshui County is the largest in physical quality grade 10, utilization quality grade 11 and economy quality grade 10. The distribution of cultivated land quality grade in Pengshui County is shown in Figure 3.

**Table 2.** Classified statistics of cultivated landquality grade area in Pengshui County.

| Grade Category | Physical Quality Grade (ha) | Proportion (%) | Utilization Quality Grade (ha) | Proportion (%) | Economy Quality Grade (ha) | Proportion (%) |
|---|---|---|---|---|---|---|
| 7 | 47.40 | 0.04 | 7.29 | 0.01 | 7.29 | 0.01 |
| 8 | 47.40 | 0.04 | 1083.19 | 0.84 | 1057.03 | 0.82 |
| 9 | 10,219.79 | 7.88 | 11,349.46 | 8.76 | 16,430.29 | 12.68 |
| 10 | 75,388.85 | 58.16 | 45,028.63 | 34.74 | 57,458.28 | 44.33 |
| 11 | 43,305.57 | 33.41 | 59,426.31 | 45.85 | 45,996.07 | 35.49 |
| 12 | 655.41 | 0.51 | 12,722.14 | 9.82 | 8668.05 | 6.69 |
| Total | 129,617.02 | 100.00 | 129,617.02 | 100.00 | 129,617.02 | 100.00 |

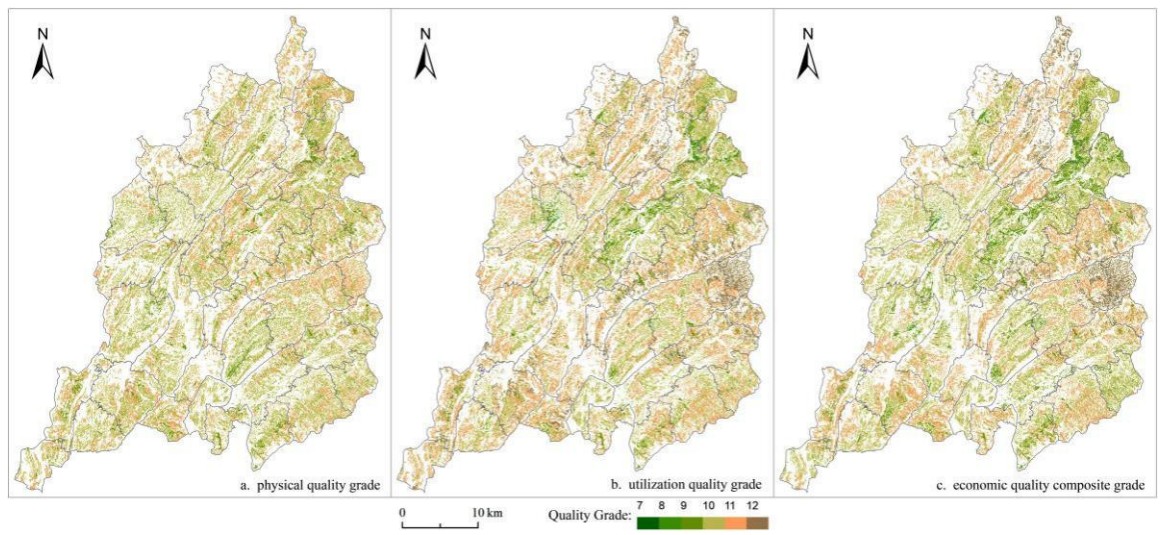

**Figure 3.** Spatial distribution map of the cultivated land quality grade of the study area.

*4.2. Spatial Auto-Correlation Analysis of Cultivated Land Quality at County Scale*

Geo Da software was used to analyze the global spatial auto-correlation of cultivated land quality in Pengshui County. The cultivated land map spot was selected as the basic unit, physical, utilization and economic of quality grade index of cultivated land were taken as the attribute values, the weight of the nearest K point was adopted (K = 4), and the calculation was carried out by referring to the spatial auto-correlation calculation formula (randomization 999 times). The results were shown (Table 3).

**Table 3.** Moran's I value of global spatial auto-correlation analysis of cultivated land quality at county level in Pengshui County.

| The Index of Category | Moran's I Value | *p* | *Z* |
|---|---|---|---|
| Physical quality grade index | 0.461639 | 0.001 * | 201.4277 |
| Utilization quality grade index | 0.594502 | 0.001 * | 262.876 |
| Economy quality grade index | 0.684261 | 0.001 * | 294.9225 |

Note: * Correlation reached significant level.

Moran's I values of the three types of cultivated land quality grade index in Pengshui County are natural, utilization, and economic of quality grade index from small to large. The results showed that the spatial dispersion of physical quality grade index, the spatial agglomeration of economic quality grade index, and the utilization quality grade index of cultivated land in Pengshui County showed significant spatial auto-correlation.

### 4.3. Spatial Auto-Correlation Analysis of Cultivated Land Quality at Township Scale

With township as the scale, cultivated land map spots as the basic unit, and cultivated land physical quality grade index, utilization quality grade index, and economy quality grade index as attribute values, the nearest *K* point weight (K = 4) was adopted and the calculation was carried out by referring to spatial auto-correlation calculation formula (randomization 999 times). The results are shown (Table 4).

**Table 4.** Moran's I value of global spatial auto-correlation analysis of township cultivated land quality in Pengshui County.

| Township | Physical Quality Grade Index | Utilization Quality Grade Index | Economy Quality Grade Index | Township | Physical Quality Grade Index | Utilization Quality Grade Index | Economy Quality Grade Index |
|---|---|---|---|---|---|---|---|
| Hanjia Town | 0.43 | 0.43 | 0.68 | Zouma Township | 0.33 | 0.26 | 0.23 |
| Baojia Town | 0.47 | 0.56 | 0.59 | Lutang Township | 0.26 | 0.62 | 0.81 |
| Yushan Town | 0.26 | 0.29 | 0.44 | Long Beach Township | 0.42 | 0.32 | 0.69 |
| High Valley Town | 0.55 | 0.39 | 0.4 | Qiaozi Township | 0.36 | 0.32 | 0.27 |
| Sangzet Town | 0.52 | 0.85 | 0.88 | Moved Joe Township | 0.37 | 0.31 | 0.27 |
| Antlers Town | 0.36 | 0.41 | 0.53 | Xintian Township | 0.76 | 0.67 | 0.73 |
| Huangjia Town | 0.30 | 0.34 | 0.48 | Meiziya Township | 0.34 | 0.26 | 0.26 |
| Puzi Town | 0.60 | 0.4 | 0.34 | Zhufo Township | 0.34 | 0.23 | 0.19 |
| Lianhu Town | 0.27 | 0.54 | 0.34 | Small Factory Township | 0.40 | 0.43 | 0.45 |
| Dragon Shot Town | 0.58 | 0.79 | 0.69 | Tonglou Township | 0.50 | 0.26 | 0.16 |
| Dianshui Township | 0.68 | 0.72 | 0.72 | Thousands of Feet Township | 0.42 | 0.51 | 0.68 |
| Yandong Township | 0.42 | 0.23 | 0.25 | Anshan Township | 0.35 | 0.55 | 0.49 |
| Luming Village | 0.59 | 0.79 | 0.84 | Compassion Township | 0.34 | 0.38 | 0.26 |
| Ping an Township | 0.58 | 0.87 | 0.90 | Shuanglong Township | 0.35 | 0.20 | 0.13 |
| Diitang Township | 0.50 | 0.31 | 0.34 | Shipan Township | 0.40 | 0.47 | 0.38 |
| Taiyuan Township | 0.46 | 0.58 | 0.65 | Dayya Township | 0.33 | 0.19 | 0.31 |
| Sanyi Township | 0.49 | 0.29 | 0.49 | Runxi Township | 0.37 | 0.21 | 0.19 |
| Union Township | 0.35 | 0.54 | 0.75 | Longxi Township | 0.39 | 0.43 | 0.38 |
| Shiliu Township | 0.45 | 0.73 | 0.82 | Zouma Township | 0.32 | 0.53 | 0.66 |
| Longxi Township | 0.31 | 0.35 | 0.33 | Average | 0.42 | 0.45 | 0.49 |
| Variable coefficient | 0.28 | 0.43 | 0.47 | Pengshui County scale | 0.46 | 0.59 | 0.68 |

As can be seen from Table 4:

(1) There is spatial consistency in cultivated land quality index. The average Moran's I values of the three indexes of cultivated land quality in each township are consistent with the Moran's I values of the global auto-correlation index at the county level, which are physical quality grade index < utilization quality grade index < economic quality grade index.

(2) There are significant differences in spatial auto-correlation among different townships. The Moran's I of cultivated land physical quality grade index was the highest in Xintian Township (0.76) and the lowest in Lutang Township (0.26). The Moran's I of cultivated land utilization quality grade index was the highest in Pingan Township (0.87) and the lowest in Dayya Township (0.19). The Moran's I value of cultivated land economic quality grade index was the highest in Pingan Township (0.76) and the lowest in Shuanglong Township (0.13).

(3) There is a great difference in the overall horizontal spatial auto-correlation between township and county. The Moran's I value of cultivated land physical quality grade index is higher than that of county level in only 12 townships, the Moran's I value of cultivated land utilization quality grade index is higher than that of county level in only 8 townships, and the Moran's I value of cultivated land economy quality grade index is higher than that of county level in only 11 townships.

(4)   There are great differences in the quality level of cultivated land in different towns. The Moran's I values of cultivated land physical, utilization, and economy quality grade index change in the same direction as those of county level, and are all greater than the Moran's I values of corresponding indexes at county level. The only three townships are Sangzhe Town, Luming Township and Pingan Township. The spatial auto-correlation of the physical quality grade index is mainly restricted by natural conditions, and the concentrated and contiguous distribution of cultivated land is the main reason why the agglomeration is higher than the county. On the basis of physical quality grade index, utilization quality grade index can be obtained by considering the utilization level, and the improvement and extension of agricultural technology can improve the difference of production capacity, so as to narrow the difference of cultivated land utilization quality. The indices such as economy take into account planting costs and benefits based on the indices such as physical and utilization, and their overall variability further weakens.

### 4.4. Spatial Auto-Correlation Analysis of Cultivated Land Quality at Village Scale

#### 4.4.1. Global Spatial Auto-Correlation Analysis

The spatial distribution of Moran's I values of the three types of cultivated land quality indexes at village level in Pengshui County is more complex than that at township level and county level (Figure 4). The minimum values of global Moran's I of the three indexes of cultivated land quality were not uniform, and the larger values appeared in Xiaoba Village of Dianshui Township. Among 301 villages (neighborhood committees) in the whole county, 7 villages (neighborhood committees) could not calculate Moran's I value or *p* value greater than 0.05, and the other 294 villages (neighborhood committees) were greater than 0. Moran's I value or *p* value is greater than 0.05 in 7 villages (neighborhood committees), less than 0 in 5 villages (neighborhood committees), and greater than 0 in the other 289 villages (neighborhood committees). Moran's I value or *p* value is greater than 0.05 in 7 villages (neighborhood committees), less than 0 in 6 villages (neighborhood committees), and greater than 0 in the other 288 villages (neighborhood committees).

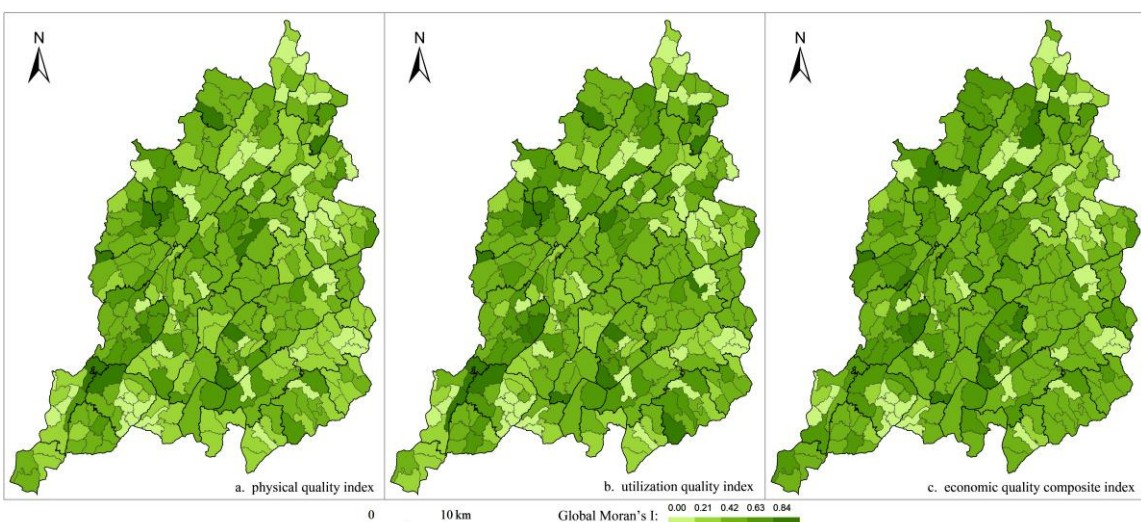

**Figure 4.** Spatial distribution of global Moran's I value of cultivated land quality index at village scale of study area.

From the point of view of spatial distribution, the spatial distribution of global Moran's I values of the three kinds of exponents is generally scattered, and pattern is not obvious.The spatial distribution of global Moran's I values of the three indices is basically the same, and the similarity between utilization and economy is higher, and the gap between them and nature is larger.

4.4.2. Local Spatial Auto-Correlation Analysis

The spatial difference of cultivated land quality mainly shows the characteristics of agglomeration and discrete distribution. In this paper, the spatial LISA distribution map is used to analyze the spatial relationship between the agglomeration and dispersion of cultivated land quality in Pengshui County. Here, NS is not significant, indicating a region with random distribution and no spatial correlation. HH and LL represent positive spatial correlation, and the region has the characteristics of agglomeration. HH represents the region with high indexes such as the quality of itself and the surrounding plots. LL type refers to the area with low index such as the mass of itself and the surrounding land. HL type and LH type represent negative spatial correlation, and the region has discrete characteristics. HL type refers to the region with a high quality index, but a low surrounding mass index. LH type refers to the area with a low quality index but a high quality index of surrounding land. With 95% confidence ($p < 0.05$), the local spatial autocorrelation results of cultivated land quality in Pengshui County are shown in Figure 5.

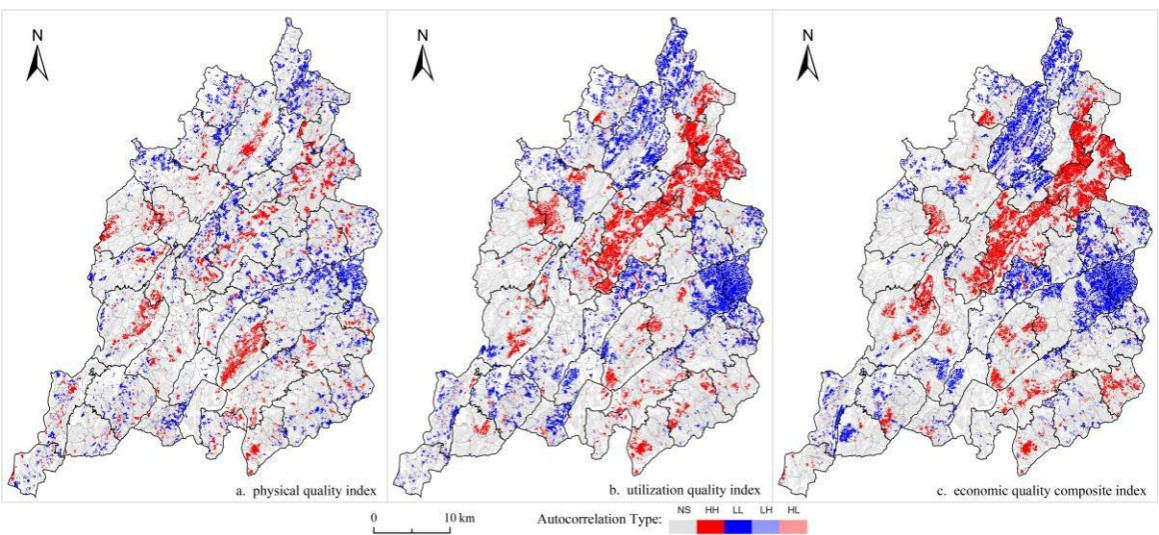

**Figure 5.** Cluster map of cultivated land quality index of study area.

Physical quality grade index: HH type was concentrated in Xintian and Baojia town in the middle of Pengshui County, Yushan in the northeast, Pingan, Luming and Dianshui in the west. LL type was concentrated in Xiaochang, Sangzhe, Zouma and Qiaozi towns in the east of Pengshui County, and in Sanyi, Lianhu and Taiyuan towns in the north. There was no obvious concentration area of HL and LH types, and they were scattered in each township.

Utilization quality grade index: HH type was mainly distributed in Baojia town in the middle of Pengshui County, Yushan in the northeast, Lianhu town and Shiliu town in the north, Pingan town and Dianshui town in the west, Sxiang town and Anzi town in the south. LL type was concentrated in Xiaochang, Sangzhe and Wanzu towns in the east of Pengshui County; Sanyi, Puzi, Taiyuan and Ditang towns in the north, and Huangjia, Runxi and Langxi towns in the south. There was no obvious concentration area of HL and LH types, and they were scattered in each township.

Economy quality grade index: HH type was concentrated in Baojia town, Xintian town and Lujiao town in the middle part of Pengshui County, Yushan town in the northeast, Lianhe town and Shiliu town in the north, Pingan town and Dianshui town in the west, Sxiang town and Meiziya town in the south. LL type was concentrated in Xiaochang and Sangzhe towns in the east of Pengshui County, Qianqiao and Wanzu towns in the middle of Pengshui County, Sanyi, Puzi, Taiyuan and Ditang towns in the north, and Huangjia, Runxi and Longtang towns in the south. There was no obvious concentration area of HL and LH types, and they were scattered in each township.

*4.5. Comparative Analysis of Spatial Auto-Correlation of Cultivated Land Quality at Different Scales*

Under different spatial scales, the spatial heterogeneity of research entities in natural environments may be enhanced or weakened [41]. In this study, different levels of administrative units were used as research scales to explore the spatial differences of cultivated land quality. It shows the changes of Moran's I values of each index at the county, township, and village level (Table 5).

**Table 5.** Moran's I value change of global spatial auto-correlation analysis of cultivated land quality under different spatial scales.

| Indicator | Spatial Scale | Physical Quality Grade Index | Utilization Quality Grade Index | Economy Quality Grade Index |
|---|---|---|---|---|
| Average | County | 0.46 | 0.59 | 0.68 |
| | Township | 0.42 | 0.45 | 0.49 |
| | Village | 0.34 | 0.30 | 0.27 |
| Variable coefficient | Township | 0.28 | 0.43 | 0.47 |
| | Village | 0.43 | 0.59 | 0.64 |
| Difference value of Moran's I | County and township | 0.04 | 0.06 | 0.19 |
| | Township and village | 0.08 | 0.15 | 0.22 |

At different spatial scales, the spatial distribution of cultivated land quality indexes of the three types in Pengshui County is similar, but there are certain differences in the same index at different scales. The specific situation is as follows:

(1) There are differences in the average Moran's I value of cultivated land quality at the different scales

Three types of index of average Moran's I values show: village level < township level < county level, the space of the cultivated land quality since the correlation with the increase in space scale increases with the increase in cultivated land plots. The main reason is that the change of cultivated land in small area is covered up by the large regional scale.

Moran's I value to the coefficient of variation of two types of indicators: village level < township level; That is, the spatial correlation of farmland quality at the rural scale is not as big as the fluctuation at the township scale.

(2) There are differences in the average Moran's I values of cultivated land quality at the same scale

The average Moran's I values of the three types of cultivated land quality indexes at county level and township level showed that: physical quality grade index < utilization quality grade index < economic quality grade index. It shows that the spatial auto-correlation of the physical index of cultivated land quality is the weakest, the utilization index is the strongest, and the economy index is the strongest. However, the average Moran's I values of three types of village level cultivated land quality indexes show that: economic quality grade index < utilization quality grade index < physical quality grade index. County level, township level and village level show different rules.

In the spatial auto-correlation analysis of cultivated land quality in Hunchun City, China, at the county level, township level and village level at the same scale, there is a completely contrary phenomenon: physical quality grade index < utilization quality grade index < economic quality grade index [24], which may be related to the topography of the research region. From the internal difference of Moran's I values of the three cultivated land quality indexes, at the township level and village level, all the cultivated land quality indexes showed physical quality grade index< utilization quality grade index < economic quality grade index. The results show that nature is the basis of utilization and economy, and priority and key tillage will strengthen the difference of utilization and output of cultivated land between regions when natural conditions are better.

## 5. Discussion

### 5.1. Parameter Selection of Auto-Correlation Spatial Weight

Quantifying the spatial agglomeration of cultivated land quality in less developed mountainous areas involves the selection of the number of adjacent spatial units of each unit in the spatial weight parameters. Both this study and the literature show that the number of adjacent spatial units is four [42]. It is better to construct the spatial weight matrix and analyze the autocorrelation analysis.

### 5.2. Impact of Scale Change on Spatial Agglomeration of Cultivated Land Quality

The agglomeration of cultivated land quality increases with the increase in spatial scale. That is, the larger the spatial scale, the stronger is the spatial autocorrelation of cultivated land quality, and the smaller its spatial variability. This conclusion is similar to that of previous studies [43]. Possible reasons: the smaller the space and the more complex the internal structure, the lower the spatial autocorrelation. In addition, the most sensitive index to spatial scale change is the natural index, followed by the utilization index, and the economic index is the least sensitive.

### 5.3. Spatial Agglomeration Types of Cultivated Land Should Be Treated Differently

The cultivated land in HH-type region should be protected, and the cultivated land in HL- and LH-type region should be transformed into HH-type cultivated land. The natural conditions of cultivated land in LL-type areas are generally poor, so active protection should be given. In case of conversions of agricultural land to construction land or other non-agricultural land, LL-type areas should be given priority [44].

## 6. Conclusions

The spatial agglomeration of cultivated land quality was studied from different administrative scales. It involves the influence of spatial weight parameter selection and scale change on the spatial agglomeration of cultivated land quality. It was found that the spatial agglomeration characteristics of township scale and cultivated land nature indices were the most stable and significant in less developed mountainous areas, which could be used as the direct basis for the zoning of cultivated land protection and the site selection of rural residents' agglomeration points. At the same time, the spatial variability of cultivated land quality and its sensitivity to spatial scale changes should be brought into the system of zoning utilization and protection of cultivated land, and the cultivated land management of different administrative levels should be treated differently, from "flood irrigation" to "precise drip irrigation". According to the natural conditions, utilization level, and output benefits, the road of land utilization and protection that conforms to the reality of underdeveloped mountainous areas is explored. Under the given natural conditions, the improvement of production technology and input level will greatly improve cultivated land utilization conditions. At the same time, the spatial agglomeration characteristics of cultivated land quality will also affect the distribution of agricultural production and agglomeration of residential areas, and then will affect the economic and social development of underdeveloped mountainous areas. In this study, the characteristics of the current spatial agglomeration of cultivated land were mainly studied from the quality index, but internal factors such as geology and geomorphology that affect the development of cultivated land were not studied. The next step will be an in-depth study on how geology and geomorphology affect the spatial agglomeration of cultivated land.

**Author Contributions:** Conceptualization, C.W. and G.L.; methodology, G.L.; software, B.W.; validation, G.L., R.L. and D.L.; formal analysis, C.W.; investigation, G.L.; resources, B.W.; data curation, D.L.; writing—original draft preparation, R.L.; writing—review and editing, G.L.; visualization, B.W.; supervision, C.W.; project administration, C.W.; funding acquisition, C.W. All authors have read and agreed to the published version of the manuscript.

**Funding:** This research was funded by National Major Water Conservancy Project of China Three Gorges Follow-up Research Project, "Research on the Key Technologies and Demonstration for the Transformation and Mechanization of Farming Fields in Chongqing" (5001022019CF50001).

**Institutional Review Board Statement:** Not applicable.

**Informed Consent Statement:** Not applicable.

**Data Availability Statement:** The new data created in this study are available on request.

**Conflicts of Interest:** The authors have declared no conflict of interest.

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
