# Peer review of "Study of the Agglomeration Characteristics of Cultivated Land in Underdeveloped Mountainous Areas Based on Spatial Auto-Correlation: A Case of Pengshui County, Chongqing, China"

_land, doi:10.3390/land11060854_

Round 1

Reviewer 1 Report

Introduction: A very detailed review especially focusing on other world Regions apart from China is needed. The aim and goals should be more clear and in detail stated.

Research Region and Data

I would have suggested to rename it “Case study”

There is a reference on cultivated land, but information regarding the crops is also needed. What type of crops, how many ha etc. Does the type of cropland affect the research results?

Fig 2 A better resolution, in order for the green color to be more vivid.

Result Analyses to be named “Results”

p. 171. Please add your source. How do you come up with these figures?

Fig 3 A better resolution, in order for the green color to be more vivid.

Conclusions

Please mention any future research implications.

Please pay attention to the text layout. The text of the abstract has to be under the affiliations. The font size of conclusions should align with the overall text.

General comments:

It is a very interesting article, and it can be a very valuable input to the agricultural research. However, the authors need to carefully look the comments and make the necessary revisions, in order for the article to be more clear, comprehensive and acceptable.

Reviewer 2 Report

The analyzed topic is very interesting and important.

 The title is adequate to the research problem being undertaken. 

The technical part of the article does not raise any objections.

The work is aesthetic.  

The correct terminology and key words was used.  

Footnotes and bibliography are correctly formulated.

 The article contains the appropriate structure. The article has been correctly divided into relevant sections, and their content coincides with their titles. However, introduction part does not explain the idea of the paper in a deep meaning and a good way. And the conclusion part and discussion part should be extended. 

The abstract does not offer enough information on the goal of the paper, methods used and main results. It should be completed to provide this information to the reader. 

The authors focused very much on the research part, but devoted far too little attention to the theoretical part.

The article uses a unsatisfactory number of references from international literature (26 items only). This is definitely not enough.

The literature references used are current and relevant with the topic.

The aim and hypothesis were not described. It is hard to refer to an article when one does not know the purpose and assumed research hypothesis.

In my opinion, there is no clearly defined purpose of the article and the research carried out. Please supplement the paper with clearly defined goals.

Why was this area of China selected as the site of analysis? I'm asking for a deeper explanation of this issue.

Round 2

Reviewer 1 Report

Dear authors,

Your manuscript has been greatly improved. However, please pay attention and correct the following:

- On discussion, highlight some how the first sentence of each paragraph after the numbers 1,2,3, in order to show that this is the title. Probably eg. Parameter selection of autocorrelation spatial weight : ......

- Sentence: (3) Treat different spatial natural related areas differently. Repetition: Different, differently. Try to ameliorate the sentence.

- Still at the conclusions, there is not a clear picture about future research on the subject. I am not asking about the benefits of your research in the future, but mostly what is lacking from this research and can be further developed.

Some sentences need to be carefully rephrased, regarding the English language. For example lines 426-429. Considering the natural conditions, utilization level and output benefit of cultivated land comprehensively, explore the way of arable land utilization and protection in accordance with the reality of underdeveloped mountainous areas.

Lines 430-431: Under the given natural conditions, the change of climate, the improvement of technology and the improvement of production level will greatly improve the
arable land utilization conditions.
Repetition of improvement, improve. The sentence needs to be rephrased.

In general, a quick look throughout the whole manuscript, of the English language is needed.

After these minor corrections the article will be ready for publishing.

Reviewer 2 Report

  Thank you very much to the Authors for taking into account my comments.
I have read the paper again and have no more comments.
I'm satisfied.
